# Evaluation of Natural Airfield Pavements Condition Based on the Airfield Pavement Condition Index (APCI)

**Mariusz Wesolowski**  **and Pawel Iwanowski *** 

Air Force Institute of Technology, 01-494 Warsaw, Poland; mariusz.wesolowski@itwl.pl
* Correspondence: pawel.iwanowski@itwl.pl

**Abstract:** Natural pavements are an important element of flights. Among other things, they allow the plane to be safely slowed down after it has exited the runway. For this reason, load bearing capacity of natural airfield pavements and strength of turf layer at a specified level are required. Currently used testing methods, such as CBR (Californian Bearing Ratio) tests or turf probe test, separately do not give a full image of pavement technical condition. The authors presented the methodology for assessing the technical condition of natural airfield surfaces based on the APCI (Airfield Pavement Condition Index). The index is based at the same time on the load bearing capacity of the surface layer up to 0.85 m and turf layer strength. The mathematical model and the classification of airfield pavements in terms of the APCI indicator are presented. The article also presents an example of using the APCI method to assess shoulders and end safety areas of the runway at one of the operating airport facilities.

**Keywords:** airfield pavement condition index; $APCI_{NN}$; natural airfield pavements; load bearing capacity; turf strength; technical condition

## 1. Introduction

### 1.1. Genesis of APCI Idea

The safety of aircraft operations within the ground maneuvering area depends on many factors. From the airport infrastructure point of view, the technical condition of the airport's functional elements (AFE) is of key importance. We are talking not only about artificial surfaces (Runways, Taxiways, or Aprons) on which aircrafts move, but also about natural surfaces (Side Shoulders, Runway End Safety Areas) providing protection in the case of an aircraft leaving the AFE. Proper management of the technical condition of pavement should be complex and include both ongoing maintenance and planning future renovations. Monitoring of the airport functional elements should be based on reliable information about the technical condition of the pavement obtained in a systemic manner. It means that each decision should be the result of collected data deriving from periodic reviews and tests. Only in this way it is possible to rationally plan repairs and renovations of all airport pavements, regardless of their structure. The experience gained by various institutions such as Cambridge Systematics, New Jersey Department of Transportation, or The Louisiana Department of Transportation and Development, shows that proper management of airport infrastructure [1], as well as road infrastructure [2–6], must rely on a detailed and up-to-date information on the technical condition of pavement.

Early diagnostics of the airport pavement's technical condition allows to avoid most of the costs incurred in the operation process, which is related to the pavement life cycle. Proper recognition of the AFE pavement technical condition at a given moment, and more importantly its anticipating in the future, is the basis for taking appropriate actions much earlier than when basing only on the assessment of the pavement in terms of visual inspection. Figure 1 shows the pavement life cycle, taking into account its technical condition. Early response to changes in the pavement technical condition allows for a significant reduction in both financial and social costs [1]. The first period of pavement

life is a slow degradation process, in which the costs of repairing any deterioration are low and the complete restoration of the element does not entail serious consequences. The dynamics of pavement degradation accelerates over the years reaching such a level that the repair costs increase dramatically year by year. During this period of time, the observed deteriorations are so large that complete pavement restoration entails costs that are incomparable to previous periods. In the short life of the surface, repair costs can increase up to five times.

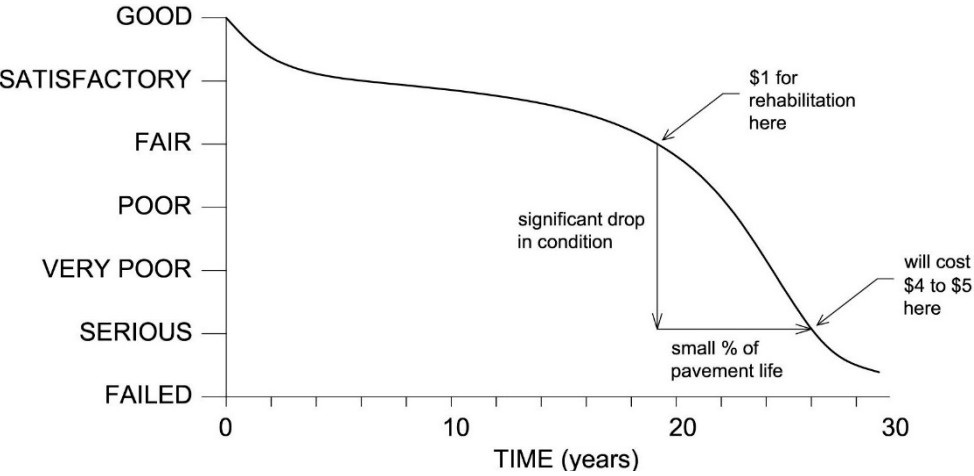

**Figure 1.** Schematic pavement life time [7].

Proper management of the pavement surfaces of airport functional elements, in particular anticipating their technical condition in the future, requires the use of appropriate tools (databases containing field test results, mathematical models for evaluation, pavement's condition prediction models, etc.) based on reliable and up-to-date information on their current technical condition. This also applies to flight protection elements such as natural airfield pavements in the immediate vicinity of the AFE.

A comprehensive assessment of the technical condition of AFE's natural pavements based on the APCI (Airfield Pavement Condition Index) is presented in the article. The APCI index is based on the results of field tests, the scope of which includes the test of the turf layer strength to a depth of 0.3 m and the test of the load bearing capacity (define by CBR) of the natural pavement to a depth of 0.85 m below the ground level. The APCI method connects two different methods into one parameter that allows clearly evaluate condition of natural pavement. APCI index describe pavement as a layer package unlike before mention methods. A further part of the work presents the assessment procedure containing the mathematical model of the indicator and its exemplary use based on real test results performed at one of the active airport facilities.

At the time of publishing this article, there are no known methods in the literature for the comprehensive assessment of the technical condition of natural airport pavements. The assessment of the technical condition of artificial airfield pavements presented in the works [8,9] is an analogy of the undertaken subject. Currently, various management systems of airport infrastructure based on various parameters of the airfield pavement are known.

### 1.2. Pavements Condition Indicators

In the airport infrastructure management, an important role is played not only by information about the current condition of the pavement, but also the ability to predict the condition of the pavement in the future. Iranian scientists in their work [5] presented attempts to develop an alternative method for determining the PCI (Pavement Condition Index), using optimization techniques based on artificial neural networks and a genetic

algorithm. The proposed approach may help in the future to reliably extrapolate the PCI index. The use of the PCI index in the estimation of Remaining Service Life (RSL) as a parameter defining the current and future condition of the airport pavement [10] may be an alternative approach. A similar approach was used in Indonesia to assess the condition of road surfaces. In the paper [11], the authors used the ANOVA method to predict RSL based on PCI. In Indianapolis, the PCI index is used as a tool to support the management of airport infrastructure. The Minimum Service Level (MSL) has been defined for individual categories of airport functional elements, which determines the time of taking corrective actions on a given element [12]. This has a significant impact not only on safety, but also on the costs incurred. Sharaf et al. [13] have shown long ago that a properly selected assessment scale for PCI indicators determines the proper maintenance of the pavement and the related costs.

Various methods of the PCI indicator determination can be found in the literature and operational practice. They are based mainly on the visual inspection of surface deteriorations. The standard method used in the world is the method using the PCI index, published in the American standards ASTM D5340 [14] and ASTM D6433 [15] described by Shahin [1]. This is a standard procedure developed and used by the US Army Corps of Engineers. Today, it is the most common test method among engineers, government agencies, and airfield pavement organizations. The PCI index is a dimensionless number from 0 to 100 describing the technical condition of the pavement, where 100 means the pavement is in perfect condition, and 0-the pavement is in a completely degraded condition, incapable of any exploitation. The PCI index is determined based on a visual inspection of pavement deteriorations. During the inspection, the type of deterioration, its number, harmfulness, and location on the selected pavement strip are taken into account. A detailed description of the PCI method is presented in monograph [16].

One of the most popular procedures derived from the PCI method is the PAVER procedure, used, e.g., by Kirbas and Karasahin [17], in order to assess the technical condition of 20 intersections in the Samsum region of Turkey. PAVER is an electronic system developed by the US Army Corps of Engineers, enabling the collection and processing of data obtained as a result of the pavement survey and its graphical representation on pavement diagrams. The Virginia Department of Transportation (VDOT) has developed the Distress Maintenance Rating (DMR), which manages pavements according to the "worst first basis". In accordance with this principle, pavement reconstruction schedules are created. The general approach to pavement assessment is based on the PCI method and the PAVER system, while VDOT has developed its own pavement condition indicators compatible with Virginia conditions [18,19].

The PCI method is based only on the visual assessment of the surface. Its main advantage is the speed of the pavement inspection, but it also has some disadvantages. The most serious of them is the inability to comprehensively assess the surface. For this reason, many researchers have undertaken to extend the method with additional parameters. An example of such a solution is the method proposed by Arhin, Williams, Ribbiso, and Anderson [20], which takes into account, apart from the visual assessment of the surface, the International Roughness Index (IRI) parameter. This parameter refers to the evenness of the surface, i.e., the properties of the surface describing the comfort of travel. Moreover, the authors presented the relationship between the IRI parameter and the PCI index, and developed a model for estimating the PCI index based on the measured IRI parameter. Scientists from Kazakhstan [21] showed their own method of assessing the pavement condition, and in addition to the IRI index, they also used the measurement of elastic deflections. In addition, they took into account macro unevennesses, ruts, and cracks. Pavement Condition Rating (PCR) was proposed as a parameter describing the condition of the pavement. On its basis, the pavement is assessed in the functional context and it is the basis for making maintenance decisions. In India [22], a method of road pavement assessment based on the Overall Pavement Condition Index (OPCI) was developed for the management of the road network of the Noida city. The OPCI model includes four

indicators: Pavement Condition Distress Index, Pavement Condition Roughness Index, Pavement Condition Structural Capacity Index, and Pavement Condition Skid Resistance Index. Each of the indicators is calculated individually, and then the OPCI is calculated based on the above indicators.

Some of the proposed procedures for assessing pavement according to the PCI index slightly automate the whole process. An example is [23], where the authors developed a weight-based approach to the PCI model. The method is based on a visual inspection of the pavement, as is the case in the traditional PCI method, while the method of determining the final index is slightly different. Each deterioration enters the model with an appropriate weight, and the entire process can be programmed. A similar approach was presented by Wesołowski, Barszcz, and Blacha [24] and Zieja, Barszcz, Blacha, and Wesołowski [25]. To assess the degree of degradation, the researchers used data obtained as a result of pavement's deteriorations and repairs inspection. Both deteriorations and repairs enter the model with a characteristic weight. The weights were established based on the harmfulness of a given deterioration or repair.

The basis for the correct assessment of the PCI index is the collection of reliable and up-to-date data on the condition of the pavement. Visual inspection is made by experts, and thus there is a human factor that can significantly affect the final result of the analysis. In order to ensure quality at the appropriate level, guides have been developed that detail how to conduct inspections and how to handle the collected data. Such a document was developed, among others for the Federal Highway Administration [4] and the Indiana Department of Transportation [12]. Italians [26] additionally proposed their own version of the deteriorations catalogue presented in ASTM D6433 [15], thus adapting it to the needs and morphology of road surfaces in Italian cities. Two types of deteriorations have been added (pits and tree roots) and a new curve of deduction as a function of deteriorations density has been added. In addition, the entire data processing cycle has been largely automated by implementing the application in the Visual Basic environment.

In order to eliminate the human factor from the data collection process, the goal is to fully automate it. In the article [3] the authors give the ARAN system as one of the examples, which enables automatic road analysis. The system is based on the data on pavement deteriorations obtained during the measurement. The data is collected in the form of photos of the pavement made with cameras and a three-dimensional image of the pavement obtained as a result of three-dimensional scanning. Unfortunately, the automatically obtained data are then processed manually by humans, which still affects the quality of information about the actual technical condition of the assessed pavement. ARAN was also used to collect distress data to find potential relations with pavement roughness and pavement condition [27].

### 1.3. Normative Documents

In civil aviation, the requirements and recommendations of international aviation organizations, such as EASA (European Union Aviation Safety Agency), ICAO (International Civil Aviation Organization) and FAA (Federal Aviation Administration), are first applied. In addition, in the national conditions, the regulations established by the Civil Aviation Authority (CAA) in this respect apply. In general, the documents, regulations, recommendations for the safe operation of the airport pavement in civil aviation includes among others:

- Annex to the Decision of the Executive Director of EASA No. 2017/021/R of December 8, 2017 implementing the fourth edition of Certification Specifications (CS) and Guideline Materials (GM) for Airport Design CS-ADR-DSN [28], 2017.
- ICAO Annex 14 to the Convention on International Civil Aviation, Airports Volume I—Aerodrome Design and Operation [29], 6th edition, 2013.
- Doc. 9157 ICAO AN/901 Aerodrome Design Manual Part 1—Runways [30], ICAO Third Edition, 2006.

- Doc. 9137 ICAO AN/898 Airport Service Manual Part 2—Pavement Surface Conditions [31], ICAO, Fourth Edition, 2002.
- Advisory Circular no: 150/5320-12C, Measurement, Construction, and Maintenance of Skid Resistant Airport Pavement Surfaces, US Department of Transportation, Federal Aviation Administration [32], (FAA) 1997 as amended.
- Guidelines No. 2 of the President of the Civil Aviation Authority of 25 January 2016 on the methods of assessing, measuring, and reporting the condition of the runway surface [33], 2016.
- ASTM D5340-12 Standard Test Method for Airport Pavement Condition Index Surveys [14], 2018.
- ASTM D6433-18 Standard Practice for Roads and Parking Lots Pavement Condition Index Surveys [15], 2018.

Similar to the known methods of assessing the technical condition of artificial airfield pavements, the authors developed an innovative model for assessing the technical condition of natural airfield pavements. The adopted model aims to support airport infrastructure managers by providing up-to-date and reliable data.

### 1.4. The Role of Natural Airfield Pavements

Air transport has undergone a huge transformation at the turn of the last three decades. The increasing take-off mass of aircraft, their construction, and continuous development resulted in the transition from airports with natural/turf to artificial surfaces. Nevertheless, this specific type of surface is still present at civil or military airports and landing zones which play a very important role in air operations. Natural airfield surfaces are responsible for ensuring safety in relation to the rejected take-off or delayed landing maneuver and possible taxing out off the runway. However, until recently, the requirements and methodology for assessing the load bearing capacity of natural airfield pavements were not regulated, which meant that they were partially neglected, and this directly affects the safety of air operations. Accidents, air incidents related to the above-mentioned situations are registered in the international base of air accidents and incidents, sometimes with very tragic consequences, and an emergency runway exit particularly dangerous in the event of inadequate load bearing capacity of the natural pavement within the runway.

According to [34], the natural airfield pavement is the airfield pavement created by proper surface preparation in order to ensure safe operation of military and civil aircraft. Currently, there are two types of natural surfaces, i.e., soil and turf surfaces [35]. Soil surfaces are made of properly prepared soil—they do not have a turf (grass) layer. On the other hand, the turf surface is a soil surface with a layer of properly selected and developed grass vegetation [36].

Depending on the type of airport and its function, natural airfield surfaces fulfil specific tasks. Currently, the natural airfield surfaces are airport functional elements, such as (Figure 2):

- emergency runway shoulders (shoulders), which protect the aircraft against damage in the event of an emergency runway exit. In case of a possible aircraft exit from the runway, natural surfaces should be prepared or constructed in order to destroy neither aircraft nor underground infrastructure, and guarantee quick restoration of the airport's operational capacity by efficient removal of the aircraft by airport services [28,29];
- runway end safety area (RESA), which guarantee that the maneuver of aborted take-off or delayed landing and possible taxing out of an aircraft from the runway can be carried out without damaging it [28,29];
- unpaved runways at smaller aerodromes or an emergency runway (AFE at military airfields) that are part of the runway intended for the take-off approach and take-off run, and for a touchdown and landing runway.

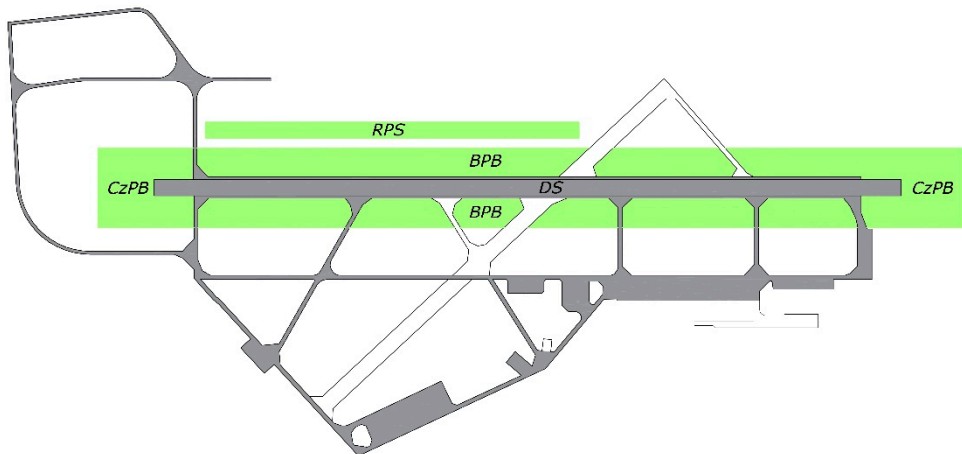

**Figure 2.** The AFE layout diagram.

International requirements for the load bearing capacity of natural airfield pavements are presented in Annex 14 to the ICAO Convention on International Civil Aviation and in Regulation (EU) No. 139/2014 of 12 February 2014 that consists of the requirements and administrative procedures for airports in accordance with Regulation (EC) No 216/2008 of the European Parliament and of the Council [37].

On the other hand, the method of assessing the load bearing capacity of natural airfield pavements and the minimum requirements for ensuring the safety of air operations by aircraft have been regulated by the provisions of the defence standard NO-17-A503: 2017 Airfield pavements. Natural airfield pavements. Load capacity testing [34]. Currently, the above-mentioned standard is used by military and civil airport services.

In order to evaluate the natural airfield pavement's technical condition of the newly built, as well as pavements already in operation, diagnostic tests should be carried out in order to determine parameters such as:

- strength of the turf layer to a depth of 0.3 m below the ground level in accordance with the requirements of the NO-17-A503: 2017 standard [34];
- testing the load bearing capacity of the natural surface to a depth of 0.85 m below the ground level in accordance with the requirements of the NO-17-A503: 2017 standard [34].

## 2. Materials and Methods

### 2.1. Inspection of Natural Pavements

In order to maintain the required load bearing capacity of natural airport pavements, and thus to ensure the safety of air operations, natural airfield pavements should be regularly maintained and repaired, i.e., all agrotechnical and biological treatments aimed at increasing the level of soil fertility and maintenance of vegetation. The basic maintenance activities of turf surfaces include [38]:

- growing and sowing mixtures of grasses; three groups of grasses are sown: tall grasses—overgrowth grasses, low grasses —underslung, legumes;
- mechanical processing of turf (harrowing, disking, rolling), aimed at brushing out the old turf, loosening the turf, removing weeds and leveling molehills, deep application of the fertilizers;
- the use of mineral and organic fertilizers;
- mowing the grass.

In order to assess the turf agricultural technology, the following qualifying features are taken into account:

- plant content (shoot density), which results from plant density, viable shoots per unit area seen from above;

- the thickness of the felt layer and its construction;
- roots density and their thickness;
- turf uniformity;
- bond strength of the turf.

In order to classify the turf, random sections of the turf are assessed in terms of thickness and root density. The uniformity and strength of the turf are also assessed.

To determine the suitability of a given soil surface for aircrafts, the rut depth *H* is of major importance [39], which is calculated according to the formula (1).

$$H = \frac{q_k^2 \cdot D}{\sigma^2 \cdot k_h} \tag{1}$$

where

$H$—rut depth [m],

$q_k$—ground pressure of one wheel of the landing gear; for the main landing gear—0.43 MPa, for the nose landing gear—0.44 MPa [kN/m$^2$],

$D$—airplane wheel diameter [m],

$\sigma$—soil strength [kN/m$^2$],

$k_h$—empirical coefficient determined from the relationship (2) [-]

$$k_h = \mathrm{m} \cdot \xi, \tag{2}$$

where

m—coefficient depending on the soil plasticity [-],

$\xi$—tire stiffness [-].

In order to quickly determine the proper load bearing capacity of the turf surface, the maximum rut depth can be assumed to be equal to 1/14 of the diameter of the main or nose wheel, as described by Formula (3).

$$H = \frac{D}{14} \tag{3}$$

The amount of deformation estimated in this way gives the belief that the aircraft, which is moving on the natural surface, will be protected from damage resulting from too intense ground movements [39].

### 2.2. The Load Bearing Capacity of Natural Airfield Pavements

Load capacity of natural airfield pavements, according to [34], is the ability of a pavement to bear a specific load from a military aircraft without the risk of damaging it, expressed by the Californian load capacity ratio CBR.

Natural airfield pavements used at the airport facilities and landing zones are subject to a comprehensive load bearing capacity assessment, the scope of which is presented in Table 1. Control tests of the load bearing capacity of natural airfield pavements should be performed periodically every 3 years [40].

At military airports and landing zones, load bearing capacity tests should be performed each time before the commencement of flight training. Then, the load bearing capacity tests are carried out on the AFE covered by the aviation training plan, and the measurements are performed before the commencement of air operations by military aircraft.

**Table 1.** Stages of a comprehensive assessment of the load bearing capacity of natural airfield pavements [34].

| | Name of Stage |
|---|---|
| 1 | Strength test of the turf layer to a depth of 0.3 m below the ground level. |
| 2 | Test of the load bearing capacity of the natural pavement layer to a depth of 0.15 m below the ground level. |
| 3 | Testing the load bearing capacity of the natural pavement layer from a depth of 0.15 m to a depth of 0.50 m below the ground level. |
| 4 | Testing the load bearing capacity of the natural pavement layer from a depth of 0.50 m to a depth of 0.85 m below the ground level. |
| 5 | Checking the condition of the subsoil up to a depth of 2.0 m below the ground level. |
| 6 | Identification of the subsoil up to a depth of 2.0 m below the ground level. |
| 7 | Analysis of the measurement results. |
| 8 | Determination of the Californian load capacity ratio CBR for the tested AFE. |

## 2.3. Strength of Turf Layer

Strength field tests of the turf layer strength are carried out in accordance with the Aerodrome Guidelines [41] using the SD turf probe, the diagram of which and the view of its practical application are shown in Figure 3.

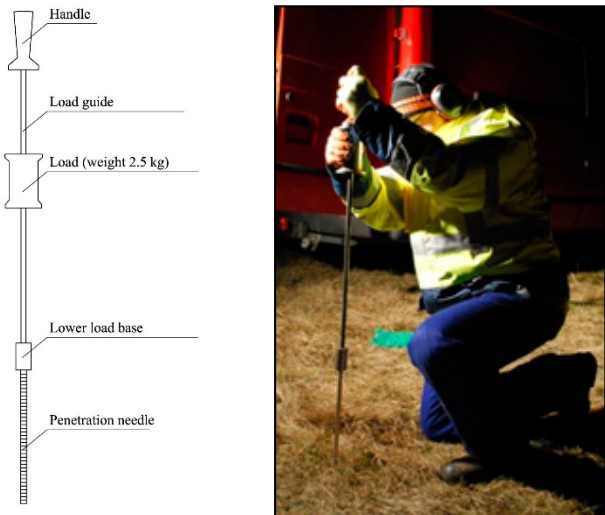

**Figure 3.** Diagram of the SD turf probe and the view of the measurements [41].

The measurement of the turf layer strength is performed down to a depth of 0.3 m below ground level and consists of measuring the number of weight strokes at depths of 0.1 m and 0.3 m by applying the probe tip into the assessed layer. From the nomograms shown in Figures 4 and 5, the values of the strength of the turf surface ($\sigma_{0.1}$, $\sigma_{0.3}$) at the depths of 0.1 m and 0.3 m, on the basis of which the average value of the strength of the layer under assessment is calculated, in accordance with the Formula (4):

$$\sigma_{\acute{s}r} = \frac{\sigma_{0.1} + \sigma_{0.3}}{2} \qquad (4)$$

where

$\sigma_{\acute{s}r}$—mean strength of the turf layer [MPa],
$\sigma_{0.1}$—strength of the turf layer at a depth of 0.1 m [MPa],
$\sigma_{0.3}$—strength of the turf layer at the depth 0.3 m [MPa].

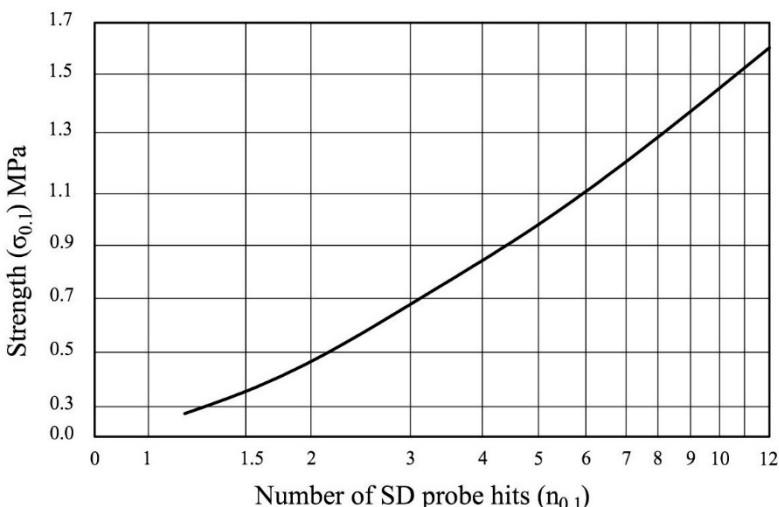

**Figure 4.** The relationship between the strength $\sigma_{0.1}$ and the number of SD probe hits ($n_{0.1}$) per the depth of 0.1 m of the probe needle drive [34].

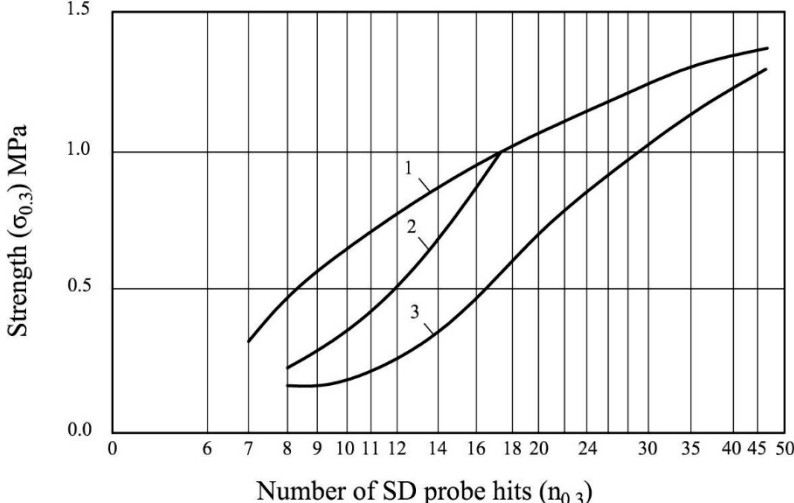

**Figure 5.** The relationship between the strength $\sigma_{0.3}$ and the number of SD probe hits ($n_{0.3}$) per the depth of 0.3 m of the probe needle drive; 1—sands, silty sands, clay sands; 2—silty sands, sandy clays; 3—mold, soils close to the optimal [34].

The average value of the strength of the turf layer for the tested AFE (determined on the basis of measurements with the SD probe) should not be less than 1.0 MPa. In exceptional situations—a single aircraft take-off or landing—the strength of the turf layer may be lower, but not less than 0.8 MPa [34].

### 2.4. Load Bearing Capacity of the Natural Pavement

The measurements of the load bearing capacity of the natural airfield pavements are carried out using the SDS probe (Figure 6, Dynamic Cone Penetrometer (DCP)), to a depth of 0.85 m below the surface, for three separate layers, i.e., 0.15 m (first layer), from a depth of 0.15 m to a depth of 0.50 m (second layer) and from a depth of 0.50 m to a depth of 0.85 m (third layer). The diagram of the separated layers of the natural airfield pavement is shown in Figure 7.

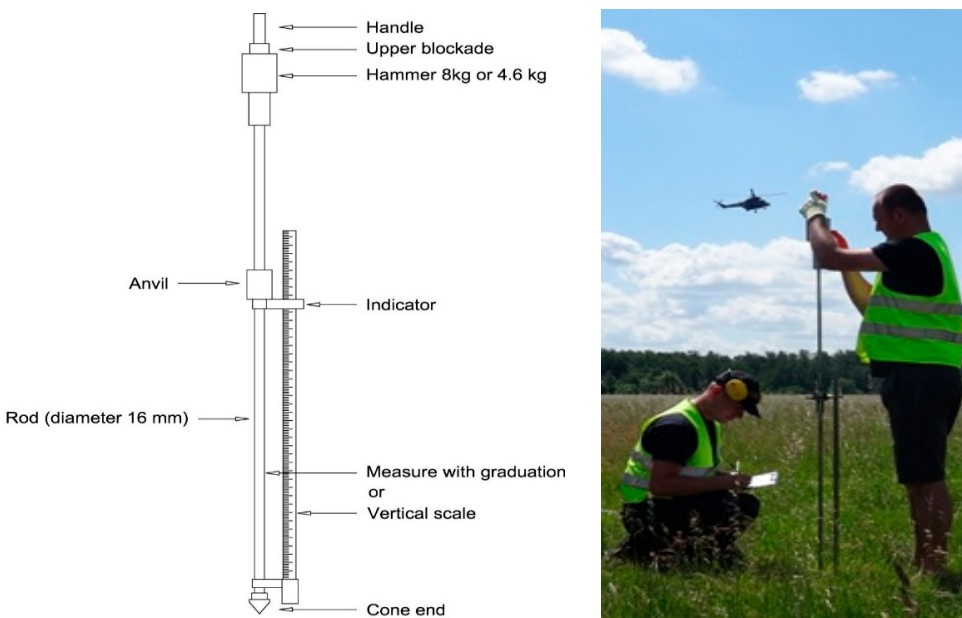

**Figure 6.** Diagram of the DCP probe and the view of the measurements [42].

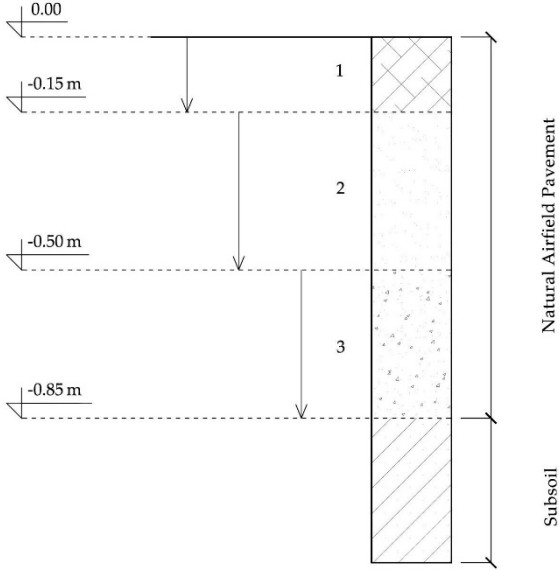

**Figure 7.** The structure of separated layers of the natural airfield pavement: 1—first layer, up to a depth of 0.15 m; 2—second layer, from the depth of 0.15 m to the depth of 0.50 m; 3—the third layer, from a depth of 0.50 m to a depth of 0.85 m [34].

Testing the load bearing capacity of natural airfield pavements with the use of an DCP probe consists of measuring the depth of the probe tip per one hit of a weight falling from a certain height [42].

The load bearing capacity of natural airfield pavements is expressed by the Californian load capacity ratio CBR, which according to [34], is calculated separately for each separated layer, in accordance with the Formula (5):

$$\text{CBR} = \frac{292}{\text{DCP}^{1.12}} \tag{5}$$

where

CBR—California Bearing Ratio [%],

DCP—probe cone depression per one stroke [mm].

The minimum value of the CBR ratio for the tested AFE should be 15% for the first layer (up to a depth of 0.15 m below ground level) and 8% for the second layer (from a depth of 0.15 m to a depth of 0.50 m below ground level) and the third layer (from a depth of 0.50 m to a depth of 0.85 m below the surface).

On the basis of years of field tests and results analysis [43–46], the evaluation criteria for natural airfield pavement's load bearing capacity were established, which are listed below in Tables 2–4.

**Table 2.** Assessment criteria of the load bearing capacity for the natural AFE to a depth of 0.15 m (layer 1).

| The Layer Strength σ (MPa) | Load Bearing Capacity CBR (%) | Natural Pavement Condition |
|---|---|---|
| σ ≥ 1.0 | CBR ≥ 15 | good |
| σ < 1.0 | CBR < 15 | bad |

**Table 3.** Assessment criteria of the load bearing capacity for the natural AFE at a depth of 0.15 m up to 0.50 m (layer 2).

| Load Bearing Capacity CBR (%) | Natural Pavement Condition |
|---|---|
| CBR ≥ 15 | good |
| 8 ≤ CBR < 15 | sufficient |
| CBR < 8 | bad |

**Table 4.** Assessment criteria of the load bearing capacity for the natural AFE at a depth of 0.50 m up to 0.85 m (layer 3).

| Load Bearing Capacity CBR (%) | Natural Pavement Condition |
|---|---|
| CBR ≥ 15 | good |
| 8 ≤ CBR < 15 | sufficient |
| CBR < 8 | bad |

Currently applicable methods for assessing the load bearing capacity of natural airfield pavements are based on point measurements. Carrying out measurements in this way requires a lot of time and the involvement of many people, and the obtained measurement results do not fully reflect the actual load bearing capacity of the pavement. The obtained results are affected with an error resulting, among others, from the fact that the measurements are performed in a discontinuous manner, i.e., every 50 m (RESA) or 100 m (emergency runway, and shoulders). The assessed natural airfield surfaces are located in the airport's operational area and access to them is limited due to the conducted air operations and working communication/navigation systems—the ILS (Instrument Landing System) for instance [47]. This means that at most airports, work related to the assessment of the load bearing capacity of natural airfield pavements can only be carried out at night after the prior introduction of the NOTAM procedure (Notice To Air Men—a concise telecommunications message containing information and provisions on the status or changes of aviation equipment, services, procedures, as well as the distress and hazards of which it is important to know in a timely manner for personnel [47,48]) and the disablement of aircraft guidance systems. However, the dynamic and resilient development of aviation, which can be observed in recent years caused, among others, by a significant increase in cheap commercial carriers and the development of modern

technologies, limits the facility's availability even at night. Therefore, the possibility of conducting field measurements is limited, which in turn extends the period of work at a facility. According to the ICAO Annual Report of the Council, in 2018 [49] the number of air operations in the world increased by 6.4% compared to 2017, and according to the forecasts of the Civil Aviation Authority (CAA), by 2035 the number of passengers using this type of transport will by more than double compared to 2018.

### 2.5. The Natural Pavement Condition Indicator APCI$_{NN}$

The APCI method for the assessment of natural airfield pavement condition is based on the results of field tests, which include:

1.  testing the strength of the turf layer to a depth of 0.3 m below ground level in accordance with the requirements of the NO-17-A503: 2017 standard [34],
2.  testing the load bearing capacity of the natural pavement to a depth of 0.85 m below ground level in accordance with the requirements of the NO-17-A503: 2017 standard [34].

The scheme of conducting the comprehensive assessment of the technical condition of natural AFE pavements is shown in Figure 8. The basis of the developed methodology is the assessment of individual input parameters based on the results of the tests and measurements carried out.

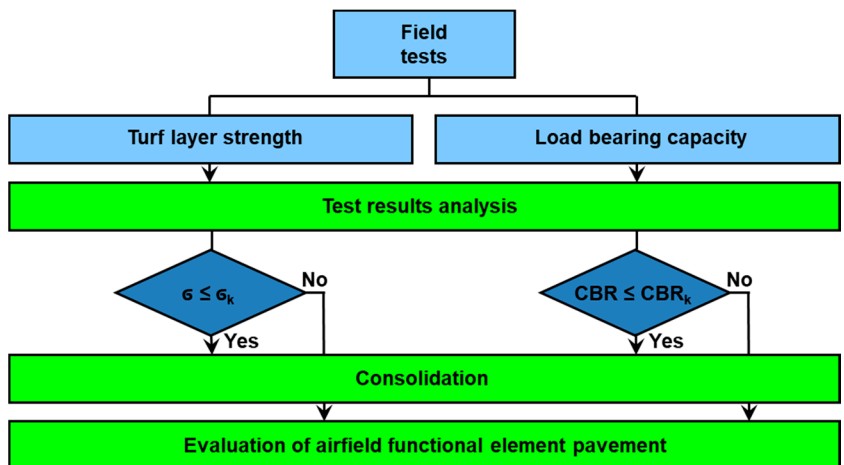

**Figure 8.** Procedure during the assessment of the technical condition of natural airfield pavements using the APCI method.

Field tests performed constitute the input data for the analysis of the process, as a result of which the output data describing the condition of the natural airfield pavement is obtained. Assessment procedure is shown in overview in Figure 9.

Data to be provided to the model of this method are derived from measurements made and field tests and the evaluation results are strength layers, and the load bearing capacity of turf surface natural made in accordance with the requirements of the NO-17-A503: 2017 standard. The test of the strength of the turf layer is performed with a turf probe to a depth of 0.3 m ppt. For each measurement point, the average strength of the assessed layer σ is calculated, which is discussed in detail in Section 2.4. On the other hand, the load bearing capacity of the natural pavement is tested with a dynamic cone probe to the depth of 0.85 m below the ground level. The parameter of the natural load bearing capacity of the airfield pavement is expressed by the Californian load bearing capacity ratio CBR. The pavement parameters collected during the field tests constitute the input data for the analysis. The developed model of the indicator APCI$_{NN}$ for natural airfield pavements is shown in the relationship (6).

$$APCI_{NN} = 100 - \frac{(w_\sigma \sigma + w_{CBR} W_{CBR})}{\sum w_i} \tag{6}$$

where

$w_\sigma$—weight for the strength of the natural surface turf layer,
$\sigma$—strength of the AFE's natural surface turf layer,
$w_{CBR}$—weight for the load bearing capacity of the natural surface,
$W_{CBR}$—average load bearing capacity index of the AFE's natural pavement,
$\sum w_i$—sum of weights.

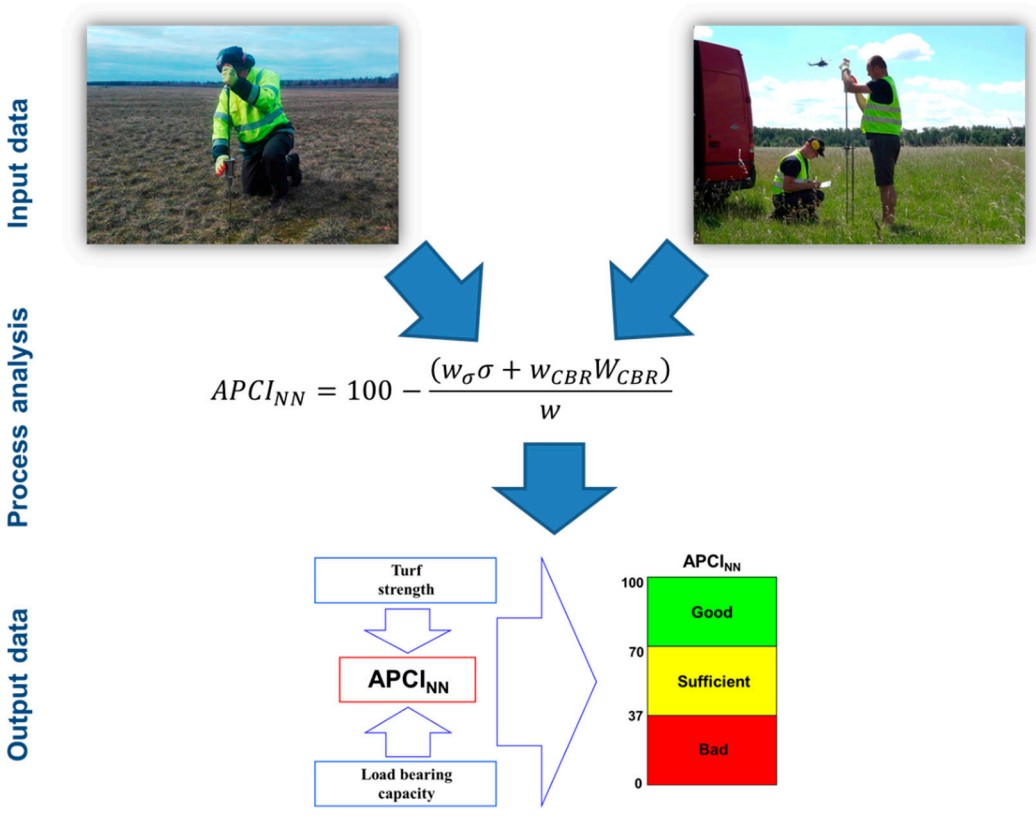

**Figure 9.** Procedure for technical evaluation of AFE natural pavement with APCI method.

Average rate $W_{CBR}$ of the AFE's natural pavement load bearing capacity is based on single results of CBR values determined in the 0–15 cm intermediate layer and in the 15–85 cm layer. It is calculated on the basis of the geometric mean of the partial load indices $W_{CBR15}$ and $W_{CBR85}$ according to the dependence (7).

$$W_{CBR} = \frac{1}{n} \sum_{i=1}^{n} \sqrt{W_{CBR15i} \times W_{CBR85i}} \tag{7}$$

where

$W_{CBR15i}$—partial load bearing capacity index of the natural pavement determined for the intermediate layer 0–15 cm in the i-th measurement point,

$W_{CBR85i}$—partial load bearing capacity index of the natural pavement determined for the intermediate layer 15–85 cm in the i-th measurement point,

$n$—number of measurement points.

Partial load bearing capacity indices are determined using the nomogram presented in Figure 10. The nomogram was created based on many years of experience of specialists in the field of airfield pavement diagnostics. For each CBR value, the value of the corresponding index is read from the vertical axis, depending on the considered natural pavement intermediate layer. The nomogram can change its form depending on the safety factor adopted $w_B$. It is assumed to be 1.2. Increasing its value results in the achievement

of lower partial indices, and thus tightening of the requirements. The CBR result in the range of $(15w_B, +\infty)$ values in each case gives a partial index of 1.0.

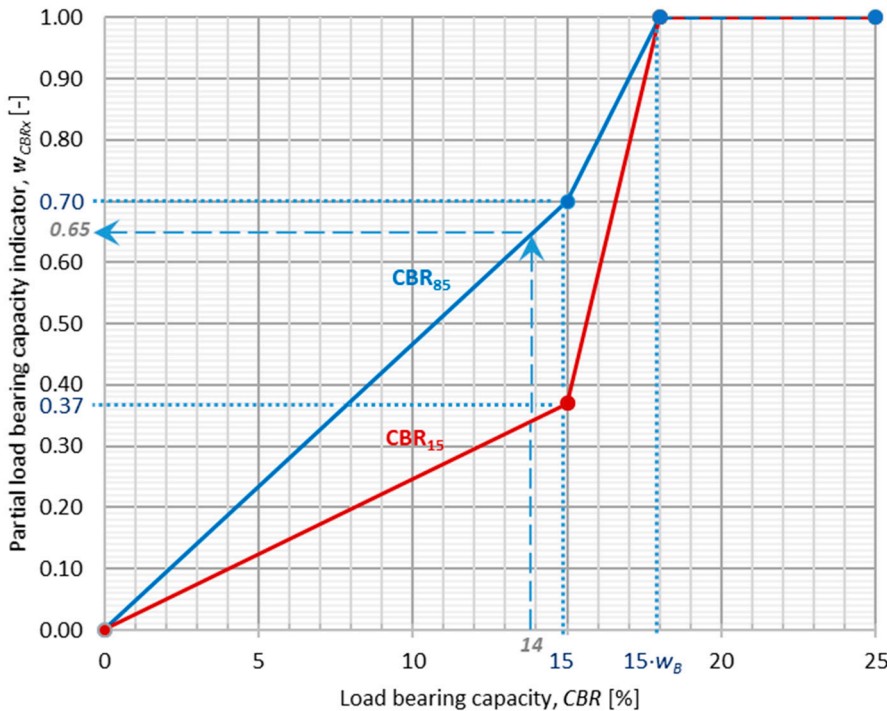

**Figure 10.** Nomogram for reading partial load bearing capacity indices of natural airfield pavements: $CBR_{15}$—curve for the intermediate layer 0–15 cm, $CBR_{85}$—curve for the intermediate layer 15–85 cm, $w_B$—safety factor ($w_B = 1.2$).

The nomogram presented above applies when the standard safety factor is 1.2. The dashed arrow shows a usage example for a CBR determined in the intermediate layer of 14%. The obtained value of the partial index $W_{CBR85}$ was in this case 0.65.

Natural airfield pavements are classified according to the criteria for assessing their technical condition, based on the obtained values $APCI_{NN}$. The criteria were developed on the basis of many years of field research.

In order to determine the scale of the values of the natural condition of airfield pavements, three categories of their technical condition assessment were introduced. The entire range of values of the selected variable, from minimum to maximum, is divided into the specified number of intervals of equal length. Those cases for which the values of the selected variable belong to one range constitute a common category. For each category, levels determining the condition of the pavement were assigned. The first category is a good level, which includes new, renovated, and exploited natural surfaces, but these surfaces will not require any scheduled renovation works in the next five years. The second category is a sufficient, intermediate level, identifies the condition of the pavement as such, in which it is justified to perform detailed tests in order to carry out treatments improving the load bearing capacity of the natural pavement. The third category is a bad level, determining immediate carrying out of load bearing capacity tests in order to determine the measures to improve the load bearing capacity of the natural pavement. Figure 11 presents the criteria for the assessment of the technical condition of AFE with a natural surface, based on the determined $APCI_{NN}$ index. Interpretations of the pavement categories are presented in detail in Table 5.

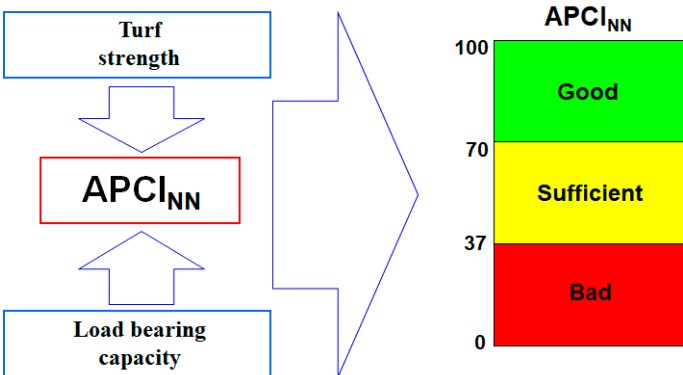

**Figure 11.** Criteria for assessing the technical condition of the AFE natural pavement using the APCI method.

**Table 5.** Criteria for assessing the technical condition of the AFE natural pavement.

| Condition | Assessment | Definition |
|---|---|---|
| Good | 71—100 | The pavement is in good technical condition, it has little or no deteriorations and requires only routine conservation works. |
| Sufficient | 38—70 | The surface is in a sufficient technical condition, it has low and medium deteriorations. Routine and major repairs need to be carried out in a short time. |
| Bad | 0—37 | The pavement is in a bad technical condition, it has highly harmful deteriorations which cause operational problems. Maintenance work should include immediate repairs and reconstructions. |

The critical value of the $APCI_{NN}$ index is the value beyond which the technical condition of the natural airfield pavement begins to deteriorate rapidly.

## 3. Results

Results of a comprehensive assessment of the technical condition of the natural AFE pavements were presented on the example of tests conducted at one of the active airport facilities in Poland. Table 6 shows the airport functional elements and their surfaces selected for the assessment.

**Table 6.** Summary of the AFEs.

| Airfield Functional Element (AFE) | Area (m$^2$) |
|---|---|
| western runway end safety area (*RESA W*) | 31,500 |
| eastern runway end safety area (*RESA E*) | 31,500 |
| northern shoulder | 318,750 |
| southern shoulder | 311,000 |

The assessment of the AFE's technical condition is made on the basis of the load bearing capacity of natural pavements tests results and the strength of the turf layer. The results of the load bearing capacity tests are presented in the form of graphs in

Figures 12–14, while the average values for each AFE are in Table 7. Charts present the results of tests performed at the active airfield. Measuring points on the shoulders were set every 100 m at a distance of about 12 m from the edge of runway, while on the RESA's they set every 50 m in the axis of runway. Each graph is assigned to one AFE. Measurements were performed in 65 points, to which the X axis on the chart corresponds.

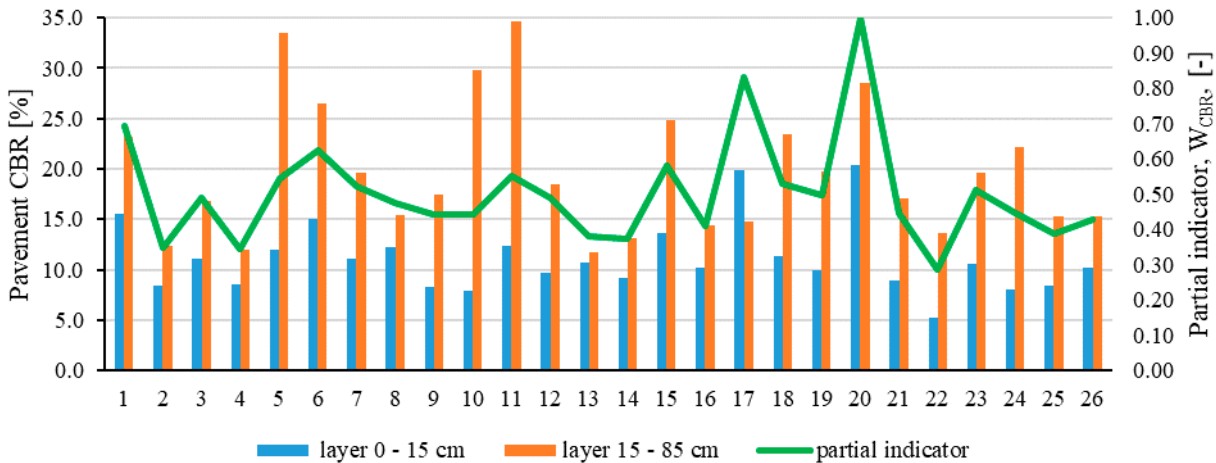

**Figure 12.** Results of the load bearing capacity of natural airfield pavements on the northern shoulder.

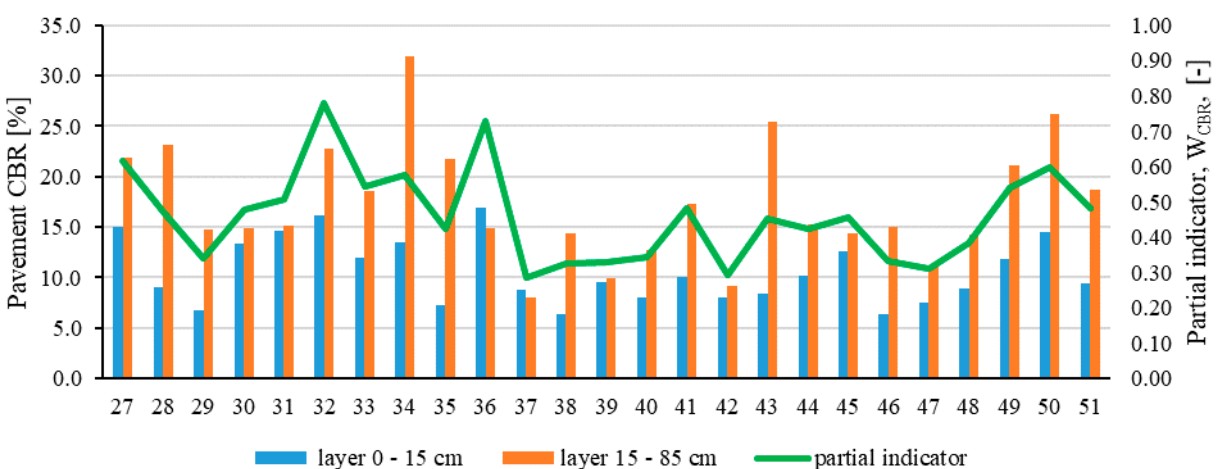

**Figure 13.** Results of the load bearing capacity of natural airfield pavements on the southern shoulder.

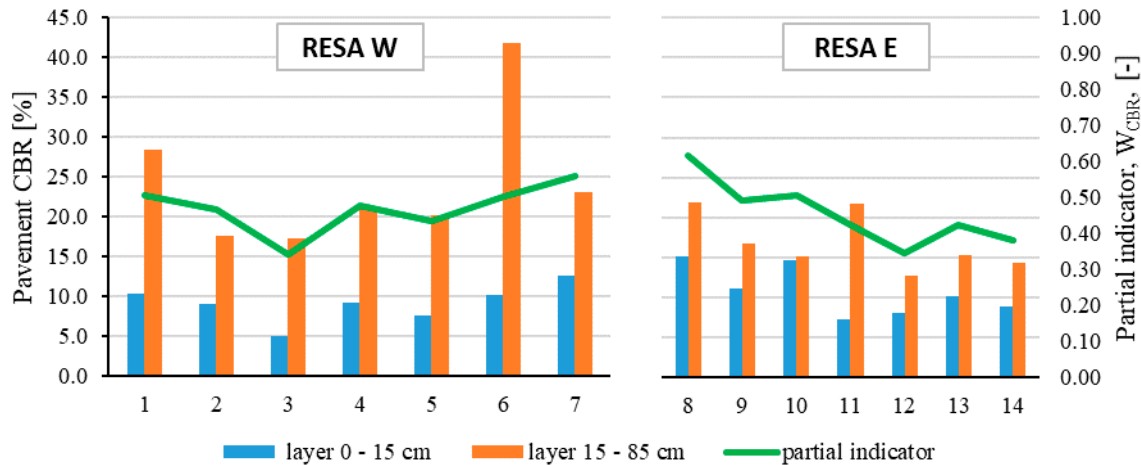

**Figure 14.** Results of the load capacity of natural airport pavements on the runway end safety areas.

**Table 7.** Average values of the load bearing capacity and the calculated CBR average of the analyzed AFE pavements.

| AFE | CBR, [%] | | $W_{CBR}$, (-) |
|---|---|---|---|
| | **Layer 0–15 cm** | **Layer 15–85 cm** | |
| RESA_W | 9.1 | 24.2 | 0.50 |
| RESA_E | 10.7 | 16.8 | 0.46 |
| northern shoulder | 11.1 | 19.7 | 0.47 |
| southern shoulder | 10.6 | 17.3 | 0.46 |

The results of the strength of the turf layer are presented in Figures 15–17. The average values of the obtained results are presented in Table 8.

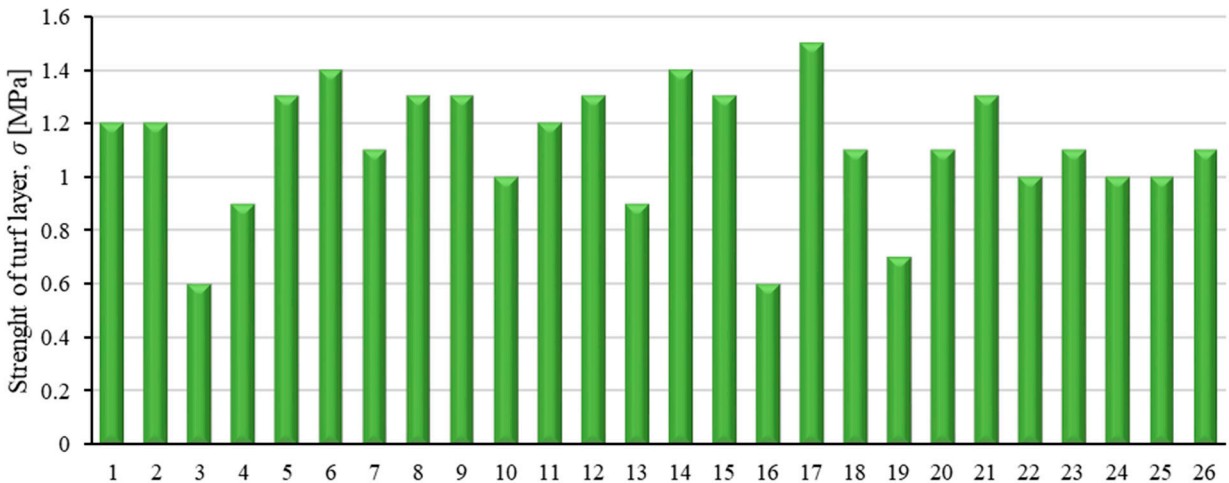

**Figure 15.** Results of the strength of the turf layer of the airfield pavement on the northern shoulder.

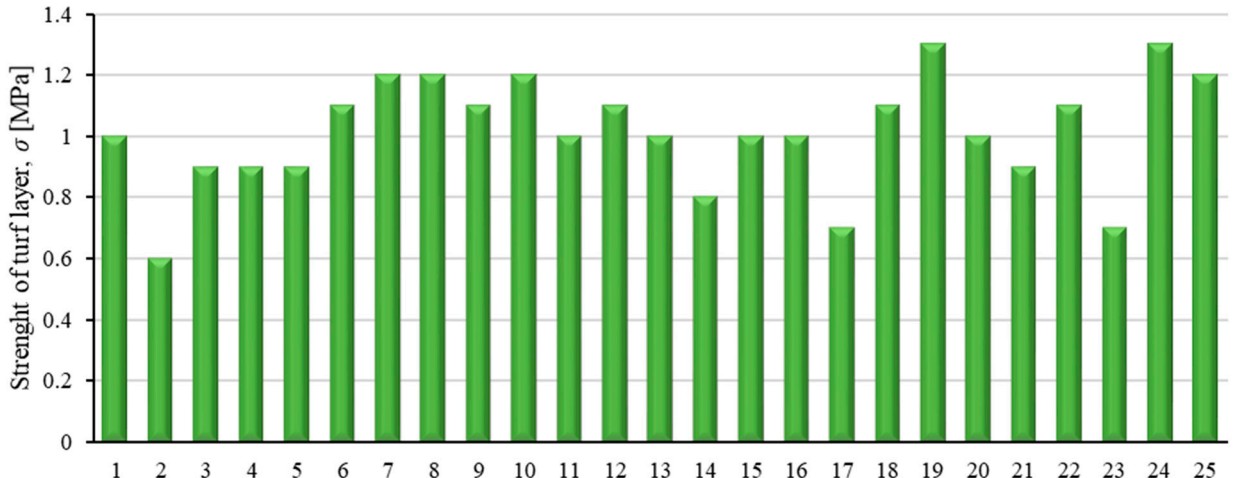

**Figure 16.** Results of the strength of the turf layer of the airfield pavement on the southern shoulder.

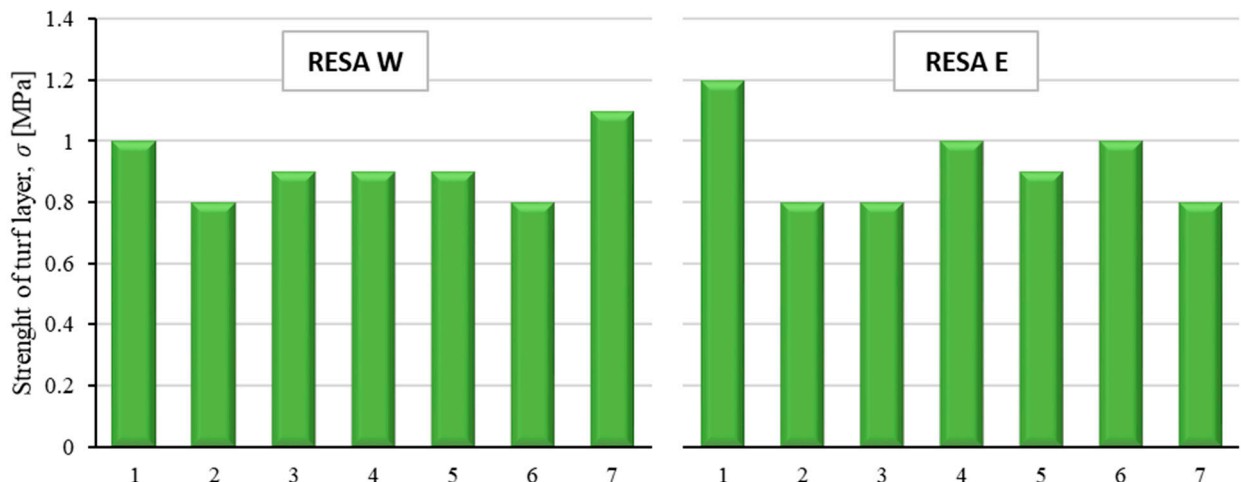

**Figure 17.** Results of the strength of the turf layer of the airfield pavement on the runway end safety areas.

**Table 8.** Average values of the strength of the AFE's turf layer.

| AFE | $\sigma$ (MPa) |
|---|---|
| RESA_W | 0.9 |
| RESA_E | 0.9 |
| northern shoulder | 1.0 |
| southern shoulder | 1.1 |

Basing on the research results of the load bearing capacity of natural pavements and the strength of the turf layer, the APCI index of AFE's natural pavements was calculated. The obtained results are presented in Table 9 and Figure 18.

**Table 9.** Results of the assessment of the technical condition of natural airfield pavements.

| AFE | $\sigma$ (MPa) | $W_{CBR}$ (-) | $APCI_{NN}$ (-) |
|---|---|---|---|
| RESA_W | 0.9 | 0.50 | 54 |
| RESA_E | 0.9 | 0.46 | 50 |
| northern shoulder | 1.0 | 0.47 | 52 |
| southern shoulder | 1.1 | 0.46 | 52 |

Basing on the obtained $APCI_{NN}$ results for the considered AFE pavements can be concluded that all surfaces are evaluated in the sufficient condition and require reinforcement. This is mostly influenced by the CBR values for the 0–15 cm layer, which significantly do not reach or slightly exceed the required value of 15%.

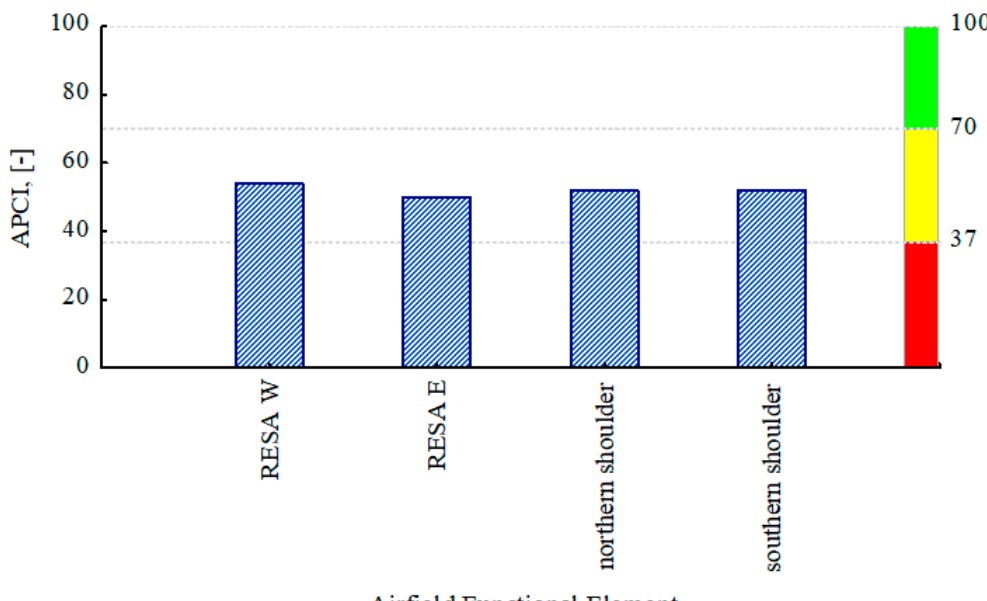

**Figure 18.** APCI$_{NN}$ values for natural AFE pavements.

## 4. Conclusions

The presented studies have been made at the airport facility. The analysis subject was the natural airfield pavements that protects flights in the immediate vicinity of the runway. The runway end safety areas and shoulders were evaluated. The CBR indicator for a layer 0–15 cm and 15–85 cm layers, as well as the strength of the turf layer to a 30 cm depth.

The CBR values for the northern shoulder fluctuated around 10–15%, the smallest value was obtained at 5%, while the highest around 35%. For the southern shoulder, the situation was reconciled similarly, however the results higher than 20% was much less and lower and that is why the lower CBR indicator values were obtained. According to the requirements for the natural pavements at airport facilities, the results were far below the required values.

Both eastern and western runway end safety areas were on the limit of requirements. As it is mentioned above, the minimum value was obtained at 5%, the highest was 20%. The exception was one point on western RESA, where a single CBR indicator for a 15–85 cm layer was 42%.

The CBR indicator values obtained from the whole airport facility was homogeneous and satisfying from the point of researchers view. It was the basis to determine that the runway vicinity is a natural pavement with a close-up parameters all around its area. Average values of the CBR indicator for end safety areas was 0.50 for western site and 0.46 for the eastern site. In the case of shoulders, it was 0.47 for the northern site and 0.46 for southern site.

In the case of the turf layer strength of the assessed natural pavements, the uniformity of the results was also at a satisfactory level. Only in five points the values were significantly lower than the obtained mean values for the whole element. The average values of the strength of the turf layer ranged around 0.9 MPa for the end safety areas and respectively 1.0 and 1.1 MPa for the northern and southern shoulder.

Basing on the results of field tests with the use of the mathematical model of the APCI index, the APCI$_{NN}$ values were obtained at the level of 52 for the shoulders and respectively 50 and 54 for the eastern and western end safety areas. The values obtained on the basis of the criteria presented in the article are in the range for pavements with a sufficient technical condition. These are surfaces with deteriorations of low and medium harmfulness. Nevertheless, routine and repair maintenance procedures should be performed in order to increase the technical condition of the surface.

Natural surfaces are an important element for the safety of air operations within the maneuvering area. They are a passive safety system and should be constantly inspected in terms of their technical condition. The requirements for natural surfaces in global aviation documents do not cover the issue in a comprehensive manner. The authors proposed a model of the $APCI_{NN}$ index, which makes it possible to evaluate the technical condition of the natural airfield pavement in a complex manner based on the results of CBR tests for the 0–15 cm and 15–85 cm layer and the results of the turf layer strength. The calculated $APCI_{NN}$ index gives clear information for the pavement manager about the pavement's technical condition. This makes it possible to take corrective actions as early as possible, which allows for savings throughout the entire life cycle of the facility.

Management of airport pavements based on reliable and up-to-date information allows not only to save financial and human resources, but also ensures the safety of air operations at the right level.

The presented $APCI_{NN}$ model will be developed in the future with parameters obtained as a result of continuous assessment of natural pavements. For this purpose, the authors undertook to develop an autonomous measurement platform that will allow the assessment of the technical condition of natural pavements along the entire length of the element, and not, as has been the case so far, only at selected measurement points (usually with an interval of 100 m).

**Author Contributions:** M.W.: Conceptualization, Methodology, Writing—Review and Editing; P.I.: Resources, Formal analysis, Writing—Original Draft and Editing. Both authors have read and agreed to the published version of the manuscript.

**Funding:** Research financed from the budget of the Ministry of Science and Higher Education as part of the statutory activity of the Airfield Division of the Air Force Institute of Technology—Project No. 0-7137-24-1-00.

**Institutional Review Board Statement:** Not applicable.

**Informed Consent Statement:** Not applicable.

**Conflicts of Interest:** The authors declare no conflict of interest.

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
