# Peer review of "Evaluation of Natural Airfield Pavements Condition Based on the Airfield Pavement Condition Index (APCI)"

_applsci, doi:10.3390/app11136139_

Round 1

Reviewer 1 Report

The basic premise of the paper is that the bearing capacity of the non-runway turf pavement should have its bearing capacity measured to meet requirements.  However, this is no evidence presented that the APCI is an improvement upon DCP or CBR.

The entire section on PCI, which is pavement distress, in irrelevant and should be deleted. 

In general, there are lots of unnecessary references and obscure references not in English

The Figures 13-19 are not explained and could result in a good paper separately.

Line 78 Indianapolis is not a state

Line 79 MSL has no context explained

Line 97 If not deleted use an English reference.

Line 120 what is dA-FEections?

Line 178  use title 150/5320-12C - Measurement, Construction, and Maintenance of Skid Resistant Airport Pavement Surfaces

Line 192 be more precise which decades

Line 201 -206  No evidence presented of inadequate bearing capacity

Equations 1, 2, and 3 all come from reference 37 which is an unexplained reference.

Line 294, and 421.  CBR is a ratio not an index. 

Line 301.  Does this require testing every day which is impracticable.

Line 360 reference to these years of testing.

This paper is based upon reference 37 but there is no context. 

Author Response

Thank you for revision. Below I attached answers for your comments.

The basic premise of the paper is that the bearing capacity of the non-runway turf pavement should have its bearing capacity measured to meet requirements.  However, this is no evidence presented that the APCI is an improvement upon DCP or CBR.

Corrected

The entire section on PCI, which is pavement distress, in irrelevant and should be deleted.

Authors think that PCI section is necessary as a genesis of the APCI method idea.

In general, there are lots of unnecessary references and obscure references not in English

Corrected

The Figures 13-19 are not explained and could result in a good paper separately.

Explanation of figures added. Data shown on figures describes field test points and authors think that it is easier for reader to analyse concrete point of data.

Line 78 Indianapolis is not a state

Corrected

Line 79 MSL has no context explained

The MSL where define as Minimum Service Life and references to Indianapolis airport infrastructure management.

Line 97 If not deleted use an English reference.

Corrected

Line 120 what is dA-FEections?

It should be “Deflections”. Corrected

Line 178  use title 150/5320-12C - Measurement, Construction, and Maintenance of Skid Resistant Airport Pavement Surfaces

Corrected

Line 192 be more precise which decades

Corrected

Line 201 -206  No evidence presented of inadequate bearing capacity

If there was proper load bearing capacity, aircraft should stop on natural pavement without any damages. In both situations nose gear collapsed into ground.

Equations 1, 2, and 3 all come from reference 37 which is an unexplained reference.

Corrected

Line 294, and 421.  CBR is a ratio not an index.

Corrected

Line 301.  Does this require testing every day which is impracticable.

These test reference to military airfields where are placed runways with natural pavement. There is requirement to test the pavement before training. You have right that this is impracticable in terms of financial but is necessary due to the flight safety.

Line 360 reference to these years of testing.

Added references to reports.

This paper is based upon reference 37 but there is no context.

Corrected

Reviewer 2 Report

Abstract

I suggest the authors to rewrite the abstract. State from the beginning of the abstract the two areas of research natural pavements and assessment of runway shoulders and safety areas and the reason why both need assessment (you already have done this partially). Be clear about the method of study. Then refer to the analysis and conclusions. The novelty of the report should be apparent from the abstract. At the current state of the abstract, the novelty is not reflected.

Line 9: define what parameters

Line 10: what testing methods to define what properties

Line 12: state the meaning of the initials APCI

Change the keywords to reflect novelty, research areas, methods, conclusions

Introduction

Generally, I found the introduction a bit general up to line 60. It raises many questions when I read it than it answers. I would suggest going through the introduction

Lines 20-23 Clarify artificial and natural surfaces. Add images or sketches or diagrams if possible.

Line 24 Clarify and rewrite “Proper management of the technical condition of pavement should be comprehensive”

Line 26 What is considered reliable information?

Line 27 What do you mean by systematic manner?

Line 29 Which institutions?

Line 37 Delete well known

Figure 1 needs more details. Add units of time and number for the x axes. 

Line 48 Which tools?

Line 54 A comprehensive assessment...

Line 62 Since they are no known comprehensive assessments give a clear definition what this means

Consider separating introduction into sections and clear titles 

The rest of the introduction after line 60 is very good.

Materials and methods & Discussion

Rearrange the two sections to demonstrate clearly materials and methods from discussion. The discussion section looks like a summary of the report and presentation of the results rather than a discussion. I think you have mixed the two sections. Reconsider

Conclusions

Good.

Author Response

Thank you for revision. Below I attached answers for your comments.

I suggest the authors to rewrite the abstract. State from the beginning of the abstract the two areas of research natural pavements and assessment of runway shoulders and safety areas and the reason why both need assessment (you already have done this partially). Be clear about the method of study. Then refer to the analysis and conclusions. The novelty of the report should be apparent from the abstract. At the current state of the abstract, the novelty is not reflected.

Line 9: define what parameters

Corrected

Line 10: what testing methods to define what properties

Corrected

Line 12: state the meaning of the initials APCI

Corrected

Change the keywords to reflect novelty, research areas, methods, conclusions

Corrected

Introduction

Generally, I found the introduction a bit general up to line 60. It raises many questions when I read it than it answers. I would suggest going through the introduction

Lines 20-23 Clarify artificial and natural surfaces. Add images or sketches or diagrams if possible.

Authors added examples of each type of surface. Sketch of natural pavement is placed on Figure 3 in section which describes role of natural pavements.

Line 24 Clarify and rewrite “Proper management of the technical condition of pavement should be comprehensive”

Corrected

Line 26 What is considered reliable information?

It reference to test results that should be correctly chosen and conducted with proper frequency. Of course tests should be conducted in accordance to normative documents.

Line 27 What do you mean by systematic manner?

Information about the surface condition should come from systematic periodic inspections and results of field tests collected in databases.

Line 29 Which institutions?

Examples were added

Line 37 Delete well known

Corrected

Figure 1 needs more details. Add units of time and number for the x axes.

Corrected

Line 48 Which tools?

databases containing field test results, mathematical models for evaluation, condition state prediction models etc.

Line 54 A comprehensive assessment...

Corrected

Line 62 Since they are no known comprehensive assessments give a clear definition what this means

Consider separating introduction into sections and clear titles

Separated into sections.

The rest of the introduction after line 60 is very good.

Thank you

Materials and methods & Discussion

Rearrange the two sections to demonstrate clearly materials and methods from discussion. The discussion section looks like a summary of the report and presentation of the results rather than a discussion. I think you have mixed the two sections. Reconsider

You have right, we deleted entire section Discussion and move it to Conclusions.

Reviewer 3 Report

the authors dealt with a very current topic. the paper is very well organized. the abastrct is clear and precise. the paper is presented well. there is only to increase the part of the discussions and conclusions. check the English language from a grammatical point of view. advisers to add some bibliographic sources.

Distress and profile data analysis for condition assessment in pavement management systems

Cafiso, S., Di Graziano, A., Goulias, D.G., D’Agostino, C.

International Journal of Pavement Research and Technology, 2019, 12(5), pp. 527–536

Author Response

Thank you for revision. I have corrected few things in manuscript. I have also added your propose of bibliograhic source.

Best regards.

Round 2

Reviewer 1 Report

I think it is necessary to state that the Pavement Condition Index (PCI) was developed only for asphaltic and cement concrete surfaces.